# Impact of Surgical Lights on the Performance of Fluorescence-Guided Surgery Systems: A Pilot Study

**DOI:** 10.3390/ani13142363

**Published:** 2023-07-20

**Authors:** Lavinia E. Chiti, Brian Park, Faustine d’Orchymont, Jason P. Holland, Mirja C. Nolff

**Affiliations:** 1Klinik für Kleintierchirurgie, Vetsuisse-Fakultät, University of Zurich, Wintherturerstrasse 260, CH-8057 Zurich, Switzerland; brian.park@uzh.ch (B.P.); mirjachristine.nolff@uzh.ch (M.C.N.); 2Department of Chemistry, University of Zurich, Winterthurerstrasse 190, CH-8057 Zurich, Switzerland; faustine.dorchymont@laposte.net (F.d.); jason.holland@chem.uzh.ch (J.P.H.)

**Keywords:** near-infrared imaging, surgical oncology, fluorescence-guided surgery

## Abstract

**Simple Summary:**

The use of near-infrared fluorophores has many promising applications in surgical oncology, both in humans and in pet animals. The reliability of the procedure is strictly related to the performances of the dedicated camera systems, which can be affected by the lighting condition of the operating theatre. In this study, we evaluated the impact of LED and halogen lights on the performances of IC-Flow^TM^ and Visionsense^TM^ VS3 Iridum by using a phantom model. Decreasing dilutions of two non-targeted (ICG, IRDye-800) and two targeted (Angiostamp, FAP-Cyan) fluorophores were imaged in a dark room, with room lights as well as LED and halogen surgical lights. The limit of detection (LOD) and mean signal-to-background ratio (mSBR) were calculated. For all of the tested dyes, the best values of LOD and mSBR were obtained in dark conditions and reasonable values were also obtained with room light conditions, while both LED and halogen lights were detrimental for the diagnostic performances of the two camera systems due to spectral contribution in the near-infrared region. When considering implementing FGS into the clinical routine, surgeons should cautiously evaluate the spectral contribution of the lights in the operating theater.

**Abstract:**

Fluorescence-guided surgery can aid in the intraoperative visualization of target tissues, with promising applications in human and veterinary surgical oncology. The aim of this study was to evaluate the performances of two fluoresce camera systems, IC-Flow^TM^ and Visionsense^TM^ VS3 Iridum, for the detection of two non-targeted (ICG and IRDye-800) and two targeted fluorophores (Angiostamp^TM^ and FAP-Cyan) under different room light conditions, including ambient light, new generation LED, and halogen artificial light sources, which are commonly used in operating theaters. Six dilutions of the fluorophores were imaged in phantom kits using the two camera systems. The limit of detection (LOD) and mean signal-to-background ratio (mSBR) were determined. The highest values of mSBR and a lower LOD were obtained in dark conditions for both systems. Under room lights, the capabilities decreased, but the mSBR remained greater than 3 (=clearly detectable signal). LOD and mSBR worsened under surgical lights for both camera systems, with a greater impact from halogen bulbs on Visionsense^TM^ VS3 Iridium and of the LED lights on IC-Flow due to a contribution of these lights in the near-infrared spectrum. When considering implementing FGS into the clinical routine, surgeons should cautiously evaluate the spectral contribution of the lights in the operating theater.

## 1. Introduction

The ability of surgeons to identify and remove all malignant tissue, both locally and at distant sites, ultimately affects the oncological outcome of cancer patients. Despite the tremendous improvements in surgical techniques and technologies, intraoperative discrimination of cancer versus normal tissue is still mostly based on surgeons’ visual and tactile assessments [1]. Fluorescence-guided surgery (FGS) can be utilized to aid in the intraoperative visualization of target tissues [2,3,4,5]. Several clinical trials have promoted the use of fluorophores, which, when excited, emit light in the near-infrared (NIR) spectrum that is then captured by dedicated camera systems [6,7,8]. NIR fluorescence imaging is increasingly being applied in the field of human and veterinary surgical oncology, with two promising indications being sentinel lymph node mapping [9,10,11] and solid tumor identification [7,12,13,14]. Briefly, for sentinel lymph node mapping, a non-targeted fluorophore, most commonly indocyanine green (ICG), is injected intradermally in four quadrants around the tumor. The dedicated camera system is then activated, and the fluorescent signal of the lymphatic vessels form the tumor to the draining lymph node can be visualized transcutaneously [9,10,11]. The fluorescent signal helps the surgeon to precisely localize and dissect the targeted lymph nodes; it can also be beneficial in guiding the dissection of a variable number of lymph nodes, as well as nodes at unpredictable sites [11]. Another great challenge of surgical oncology is discriminating diseased from healthy tissue during surgical dissection. NIRF fluorophores can be administered systemically and can accumulate in the neoplastic tissue, hence aiding the surgeon in visualizing the tissues that must be removed and potentially increasing the chances of achieving microscopically tumor-free surgical excision [13,14].

Indocyanine green was the first fluorophore to receive approval for clinical use; thus most camera systems have been developed to detect ICG [15]. Unfortunately, ICG is a non-selective, non-targeted fluorophore, and while it works well for sentinel lymph node mapping and NIR angiography, its potential to target neoplastic tissue for visualization during resection is limited [6,9,10]. Hence, there is growing interest in the development of targeted molecular dyes consisting of NIR fluorophores conjugated to target specific ligands that can bind the tissues of interest [12,16]. Non-selective fluorophores, such as ICG, are speculated to accumulate in tumor tissue due to the so-called “enhanced permeability and retention effect”, which was described by Matsumura and Maeda (1986) [17]. Following this theory, the blood capillaries within the tumor tissue have defective endothelial cells with wider fenestrations, which promote the passage and accumulation of small molecules such as non-targeted fluorophores. Due to this mechanism, non-targeted fluorophores can potentially accumulate in any tissue with increased capillary permeability, for example, inflamed tissue, thus compromising the reliable identification of neoplastic tissue [17,18]. Conversely, targeted fluorophores are specifically designed to bind molecules that are over-expressed by tumor cells, hence allowing for more specific discrimination between neoplastic and non-neoplastic tissue [12,16].

While ICG is normally imaged at micromolar concentrations, these molecular probes accumulate in the target tissue in medium to low nanomolar concentrations, thus making detection more challenging. The low tissue concentration combined with the fact that most camera systems are optimized for ICG, which has a slightly different spectrum than targeted dyes, represent a challenge. Given that intraoperative decision-making relies on the ability of the camera system to detect the fluorescent signal emitted from the fluorophore, it is crucial to assess the ability of the available systems to identify the signals from different fluorophores at different tissue concentrations [19,20]. The comparison of the performance of commercially available imaging systems with different dyes is also crucial to providing a standard reference for the interpretation of results from different clinical trials, and ultimately to supporting the rapid diffusion of FGS. In addition, it is mandatory to allow surgeons to understand the benefits and limitations of each system, as well as to choose the correct imaging system for the intended indication.

A recent review defined standard criteria for evaluating the performance of imaging systems for NIR detection [15]. An important feature of an open fluorescent imaging system is the ability to operate under ambient light conditions in a surgical theater with a reasonable signal-to-background ratio [15]. While modern imaging systems offer this option, tungsten and halogen bulbs can significantly impair the sensitivity of an imaging system given their light emission spectrum, which is in the 600 to 850 nm wavelength range. In contrast, new-generation light sources such as LED and compact fluorescent lights have minimal output in the infrared spectrum, and, therefore, might be beneficial when using NIR imaging [15,21]. However, the actual impact of surgical lights on the performance of open NIR camera systems has received little attention [21].

The aim of this study was to evaluate and compare the performances of two commercially available imaging systems designed for open FGS: the IC-Flow^TM^ and Visionsense^TM^ VS3 Iridum. Their performances were assessed by evaluating the detection of two non-targeted (ICG and IRDye-800) and two targeted fluorophores (Angiostamp^TM^ and a newly synthetized Cyan dye targeting the fibroblast-activating protein FAP) under different room light conditions, including ambient light, new generation LED, and halogen artificial light sources, which are commonly used in operating theaters. We hypothesized that both imaging systems would be able to detect the untargeted as well as the targeted dyes. Furthermore, we hypothesized that the capabilities of the commercially available imaging systems would change significantly based on the type and spectrum of the light source, with a greater impact on the IC-Flow^TM^ system and a lower impact on ICG compared to other fluorophores.

## 2. Materials and Methods

The testing process included the evaluation of two non-targeted fluorophores, ICG and IRDye-800, and two targeted NIR fluorophores, Angiostamp^TM^ targeting α_v_β_3_ integrins and FAP-Cyan, a newly synthetized molecular probe targeting the fibroblast activating protein (FAP). The chemical formula and absorption/emission spectra of these fluorophores have been described in a previous publication [22].

Each fluorophore was diluted in phosphate-buffered solution to reach the following concentrations: 10 micromolar, 1 micromolar, 0.1 micromolar, 10 nanomolar, 1 nanomolar, and 0.1 nanomolar. Sixteen phantom kits were set for the experiment. Each phantom kit consisted of a square-shaped phantom holder measuring 10 × 10 × 2.2 cm, which was 3D-printed in black rigid polyurethane, then mixed with alcohol soluble nigrosine and TiO_2_ particles to increase background absorption and reduce scattering. The phantom holders were designed to accommodate 4 Eppendorf tubes, with windows for each Eppendorf tube to allow for detection at each corner; 3 Eppendorf tubes were filled with decreasing dilutions of a fluorophore and 1 Eppendorf tube contained only the buffer solution (control). For each fluorophore, two phantoms were assembled: one with the three micromolar dilutions plus the control and one with the three nanomolar dilutions plus the control (Figure 1).

The obtained phantoms were stored in dark conditions at a temperature of −18 °C. On the day of image acquisition, the phantoms were allowed to equilibrate at room temperature for 2 h in a dark environment.

Image acquisition was performed using two camera systems: IC-Flow^TM^ (excitation peak at 740 nm; detection peak: 830 nm) and Visionsense^TM^ VS3 Iridum (excitation peak: 805 nm (laser); detection range: 825–860 nm). To evaluate the impact of varying light conditions on the sensitivity of each camera towards each fluorophore, the phantoms were imaged with both camera systems under the following conditions:
-Dark room conditions, i.e., with no artificial or natural light sources;-Room light conditions, i.e., with the room lights turned on, but the surgical lights turned off;-Surgical light type LED (Simeon Highline Sim, LED 7000, Simeon Medical GmbH & Co. KG, Tuttlingen, Germany), i.e., with the room lights and surgical lights with LED bulbs turned on;-Surgical light type halogen (MACH M3 Decknmodell, Dr. Mach GmbH & Co. KG, Grafing, Germany), i.e., with the room lights and surgical lights with halogen bulbs turned on.


The emission spectra of each surgical light were measured to assess the interference with the NIR spectrum (Figure 2). When the phantoms were imaged with Visionsense^TM^ VS3 Iridum, a working distance of 30 cm between the camera and the phantom was maintained as per the manufacturer’s recommendations; the shutter was set at 1961 and 176; and the gains were set at 2427 and 100. For the IC-Flow^TM^, a standard working distance of 20 cm was chosen, which was in line with the manufacturer’s recommendations. The light and intensity were set at 100%.

The obtained images were processed using open-source software (ImageJ). The regions of interest (ROI) of 30 × 30 pixels were manually drawn and centered over each window (including controls) and on the center of the phantom holder, which was used as the background. The signal-to-background ratio (SBR) and limit of detection (LOD) were calculated as measures of sensitivity for each fluorophore imaged with each camera system under each lighting condition. The SBR was obtained by dividing the mean signal of the ROI on the area containing the dilution of interest by the mean signal of the ROI on the background [23]. The LOD was defined as the smallest dilution with a SBR ≥ 1.5 [24].

The values for SBR were documented using median, mean, range, standard deviation, minimum, and maximum.

To evaluate the impact of various lighting conditions on the performances of each camera system, pairwise comparisons of the mean values were made, considering the SBR as the dependent variable. Significance was set at *p* = 0.05. Statistical analysis was performed with IBM SPSS for Macintosh, version 28.0.

## 3. Results

### 3.1. Limit of Detection (LOD)

The limits of detection for each fluorophore imaged with each camera system under the four different lighting conditions are listed in Table 1.

When the phantoms were imaged using the Visionsense^TM^ VS3 Iridum in dark conditions, the LOD was 1 μmol for IRDye-800 and FAP-Cyan and 10 nmol for ICG and Angiostamp^TM^. Under room light conditions, the LOD was improved to 10 nmol for IRDye-800 and remained unchanged at this dilution for Angiostamp^TM^, while it reached 0.1 nmol for ICG and 0.1 μmol for FAP-Cyan. With this camera system, the LOD under LED lighting conditions was 0.1 μmol for IRDye-800 and Angiostamp^TM^, 10 nmol for ICG, and 1 μmol for FAP-Cyan. With the halogen bulbs, the LOD was 1 nmol for all tested fluorophores.

When IC-Flow^TM^ was tested in dark conditions, the LOD was 0.1 μmol for IRDye-800, ICG, FAP-Cyan, and Angiostamp^TM^, while it reached 10 nmol for ICG. The performances of this camera system with the room light remained unchanged for all fluorophores except for ICG, for which LOD was worse by one dilution (0.1 μmol). When tested under the LED lighting conditions, IC-Flow^TM^ resulted in an unchanged LOD of 0.1 μmol for ICG and FAP-Cyan, while the LOD was reduced to 1 μmol for IRDye-800 and Angiostamp^TM^. With the halogen bulbs, the LOD remained at 0.1 μmol for ICG, FAP-Cyan, and Angiostamp^TM^ imaged with IC-Flow^TM^, while it was 1 μmol for IRDye-800.

### 3.2. Impact of Lighting Condition on Signal-to-Background Ratio (SBR)

The mean values of SBR, along with the range and standard deviation for each camera system with each tested lighting condition, are reported in Table 2.

The mean overall SBR for the Visionsense system was at its highest in dark conditions. The mean SBR decreased with surgical lights, but with this system, the lowest value was found for the halogen bulbs. For the Visionsense system, the mean SBR differed statistically between the dark conditions and the halogen bulbs (*p* = 0.045).

When considering each fluorophore which was separately imaged with Visionsense, IRDye-800, FAP-Cyan, and Angiostamp showed the same pattern: the mean SBR values were higher in dark conditions, followed by room light. SBR decreased when the surgical lights were turned on, with similar values for halogen bulbs and LED lights. For ICG, conversely, the highest SBR was recorded with room light, followed by LED light and dark conditions, and it decreased with the halogen bulbs. Reflection artifacts were recorded in dark conditions due to the high signal of the highest concentration of ICG, while diffuse fluorescence artefacts occurred when the phantoms were imaged with LED lights (Figure 3).

The mean overall SBR for the IC-Flow system was the highest in dark conditions, followed by room light conditions. The mean SBR decreased with surgical lights, with the lowest value recorded for the LED lights.

When considering each fluorophore separately imaged with IC-Flow, the same trend was noticed for all of them: the highest SBR were recorded in dark conditions, followed by room light, while the lowest values occurred when the images were captured with the LED lights turned on. None of the differences in SBR were of statistical significance.

## 4. Discussion

Near-infrared imaging is an emerging field in veterinary and human surgical oncology. As interest in the application of NIR rises and different manufacturers develop more and more clinical imaging systems, it will become important to identify the strengths and limitations of each system. Among various technical specifications, the following questions are especially important from a surgeon’s perspective:
(1)The surgeon needs to understand whether the purchased system can only image the licensed ICG, or if targeted imaging is also generally feasible with other NIR fluorophores.(2)The surgeon needs to understand whether special prerequisites (such as specific surgical lighting) must be considered when setting up an operating room (OR) where open NIR imaging is planned.


In the present study, we demonstrated that different types of surgical lights, including new-generation LED lights, impact the performance of the two tested fluorescent imaging systems by reducing the capability of these imaging systems to detect lower concentrations of fluorophores and impairing the signal-to-background ratio. As is consistent with previous reports, regardless of the lighting conditions, the best performances in terms of SBR and LOD were obtained for the Visionsense System, and both systems performed better when imaging ICG compared to the other fluorophores [15].

The field of FGS, based on NIRF fluorophores, has experienced great development in recent years, in both human and veterinary medicine [1,8,10,11,25]. The augmentation of fluorescent imaging has the potential to improve the ability of surgeons to detect diseased tissues and spare healthy ones [14,18]. However, the magnitude of the clinical benefit ultimately depends on the capability of the employed camera system to detect the fluorophore of interest in clinical conditions. Models of several phantoms have been developed and validated to test the performance capabilities of fluorescent camera systems and to determine the effects of variables such as tissue composition and depth on the observed fluorescent signal [26,27,28]. Although it is accepted that the light conditions of the operating theater affect the performances of fluorescent camera systems [15,29,30], benchmarking of camera systems with phantom models has previously been performed in standard dark conditions only.

In a recent review, it was mentioned that surgical lights can impair the performances of fluorescent camera systems because they emit light in or close to the NIR spectrum, which can, therefore, interfere with the light emitted by the fluorophore [15]. On the contrary, the interference of room lights should be lesser, and in the same review, it was stated that the ability to operate with ambient room lights tuned on is a desirable feature of open fluorescence camera systems [15]. In our study, the highest values of SBR and lower LODs were obtained in dark conditions for both camera systems, as was expected. With the room lights turned on, the capabilities of the systems decreased, but the mean recorded values of SBR remained greater than 3, which is the reported cut off for a clearly detectable signal [24,31]. These results suggest that both tested cameras performed well under room light conditions, confirming their suitability for clinical use. Sophisticated strategies have been proposed to reduce background contamination, such as pulsing of the excitation light source or frequency modulation and lock-in detection; systems that embed these features and are able to perform some sort of background correction are thought to perform better than those that cannot [29,30]. It is, however, interesting to notice that, although neither camera system tested in the present study performed background correction [15], they are still suitable for usage with room lights turned on, providing satisfactory SBR values.

When surgical lights were turned on, a trend towards a worsening of LOD and SBR was observed for both camera systems, with a greater impact from halogen bulbs on the performance of Visionsense^TM^ VS3 Iridium and from the LED lights on the performance of IC-Flow. This result is partially in contrast with what was reported by Dsouza and coworkers, who suggested that newer LED lights should have a lesser impact on the fluorescence signal quality [15]. This difference can be explained by the fact that our LED lights had a significant spectral contribution, causing interference in the emission spectrum of the NIR fluorophores, especially when the IC-Flow system was used.

When considering only the ICG imaged by Visionsense, the mean SBR was lower in dark conditions compared to when the room lights or LED lights were turned on. It should be noted, however, that the operator who processed these sets of images reported the occurrence of artifacts of reflection in dark conditions, and of fluorescence artefacts with the LED lights on. In clinical reality, both artefacts interfered with the true signal and hampered the correct identification of the target.

When considering all the fluorophores, the values of SBR for the Visionsense were higher in dark conditions, slightly decreased with room light, and clearly decreased with LED lights. The targeted fluorophores have promising applications in FGS, given their ability to selectively bond molecules of interest, thus concentrating the fluorescent signal in specific tissues [8,12,16]. Given the low concentration at which molecular probes are imaged, the negative impact of surgical lights on their detection is even more concerning. In the study presented herein, we tested two targeted fluorophores: the Angiostamp and FAP-Cyan. Under room light conditions, the Visionsense system was able to detect nanomolar concentrations of Angiostamp, while the LOD for FAP-Cyan was restricted to micromolar concentrations. The LOD for this system with halogen light was also in the nanomolar range, while it was confined to the micromolar range with the LED lights. These results suggest the ability of the Visionsense system to operate with room lights for the detection of some molecular probes. On the contrary, IC-Flow did not allow for the detection of nanomolar concentrations of the targeted fluorophores under any light conditions, posing questions regarding on the suitability of this system for application with target dyes and room light conditions.

We confirmed our hypothesis that surgical lights have a considerable impact on NIR imaging, and that these impacts vary with different bulbs. Notably, the Visionsense system had already been tested in another study, which solely evaluated the limits of detection of different open imaging systems. In that study, the system outperformed all other imaging systems. Thus, the current study included a system that had already been confirmed to be one of the most sensitive imaging systems currently available, and still, we documented considerable impacts of lighting and dye composition on the system. Currently, many manufacturers offer NIR systems, including systems that have been originally designed for endoscopic imaging. While not designed for open applications, several studies have reported their use in open settings. Unfortunately, we were not able to include any of these systems in our testing, but it must be anticipated that these systems are most likely even more prone to artefacts and impacts of lighting that the imaging systems tested in this study. However, as we were not able to test any of these systems, this argument needs to be tested in further studies to validate whether endoscopic systems really represent a valid option for open NIR imaging. Until then, surgeons need to be aware of these potential limitations.

The present study has several limitations, mainly related to the fact that images were acquired only once for each phantom, thus limiting the number of measurements obtained and lowering the power of the statistical analysis. This could explain the lack of significance for most of the comparisons. Due to the intrinsic limitations of this study, our results should be considered to be primary, and should trigger further systematic investigations into the impacts of different types of surgical lights on the performances of various commercially available NIRF camera systems. This will be useful in drawing more detailed, final conclusions on this topic that can aid the surgeons in proper decision-making when implementing FGS.

Future studies should also focus on developing experimental models that are suitable for testing the performance capabilities of fluorescence camera systems, in dark and various lighting conditions, to simulate real-life conditions and to provide the end user—the surgeon—with practical information on their usage in the operating theater. Recently, some manufacturers have also developed surgical lights specifically designed to allow for open NIR imaging under normal surgical conditions with surgical light. Future studies are needed in order to test whether these lights can actually be used without impacting the camera systems.

## 5. Conclusions

In conclusion, in the present study, we confirmed that the two tested camera systems had reasonable sensitivity under room light conditions, while surgical lights had a negative impact on their performances. For the Visionsense system, improved LOD values were recorded under dark and room light conditions for all fluorophores, and this value was most negatively affected by new-generation LED lights. Similarly, with the IC-Flow system, the best values of LOD were obtained under dark and room light conditions, but this value was negatively impacted by halogen bulbs and LED lights, especially when the targeted fluorophores were imaged. When considering implementing FGS into the clinical routine, surgeons should cautiously evaluate the spectral contribution of the lights of the operating theater.

## Figures and Tables

**Figure 1 animals-13-02363-f001:**
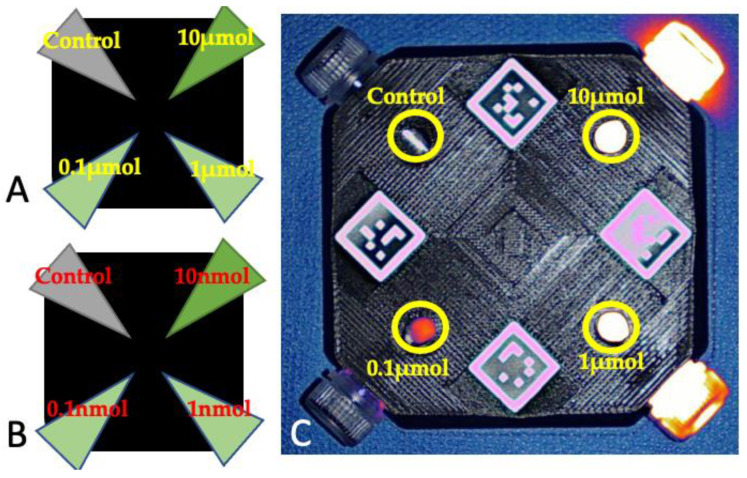
Schematic representation of the phantoms containing the micromolar (**A**) and nanomolar (**B**) dilutions of the fluorophores. (**C**) Phantom containing the micromolar dilutions of ICG, imaged with Visionsense under room light conditions. The areas where the dilutions were imaged are marked with yellow circles and the corresponding dilution.

**Figure 2 animals-13-02363-f002:**
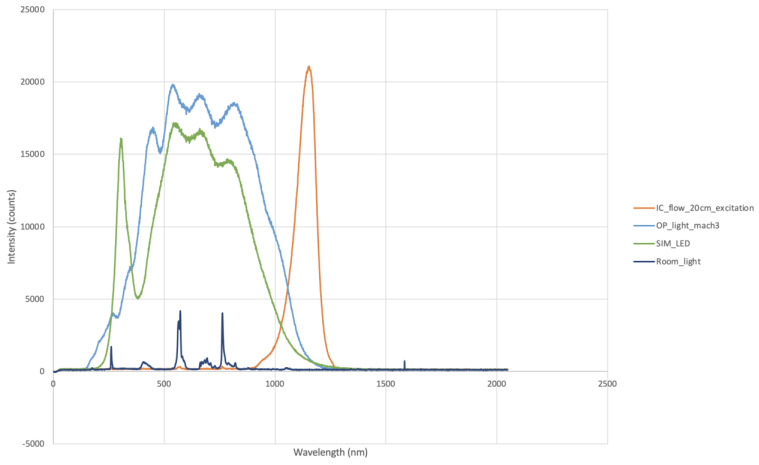
Emission spectra of LED lights (SIM_LED), halogen bulbs (OP_light_mach3), and room lights (Room_light). Note the contributions of both types of surgical light in the near-infrared spectrum.

**Figure 3 animals-13-02363-f003:**
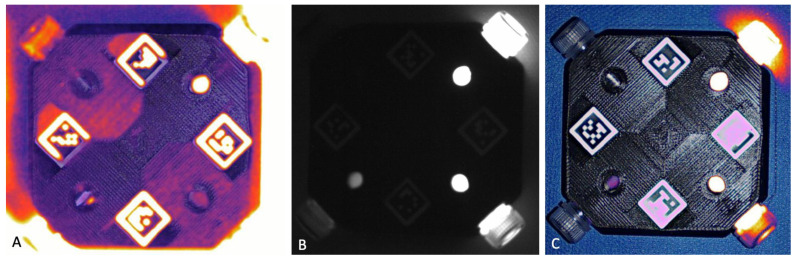
Diffuse fluorescence recorded by Visionsense due to interference of the LED light (**A**) in comparison with signal quality in dark conditions (**B**) and room light conditions (**C**).

**Table 1 animals-13-02363-t001:** Limit of detection for each fluorophore imaged with the two camera systems (Visionsense^TM^ VS3 Iridium and IC-Flow^TM^) under varying lighting conditions. ICG: indocyanine green.

Camera System and Lighting Condition	Fluorophore	Limit of Detection (LOD)
Visionsense^TM^ VS3 Iridum—DARK	IRDye-800	1 μmol
	ICG	10 nmol
	FAP-Cyan	1 μmol
	Angiostamp^TM^	10 nmol
Visionsense^TM^ VS3 Iridum—Room light	IRDye-800	10 nmol
	ICG	0.1 nmol
	FAP-Cyan	0.1 μmol
	Angiostamp^TM^	10 nmol
Visionsense^TM^ VS3 Iridum—LED light	IRDye-800	0.1 μmol
	ICG	10 nmol
	FAP-Cyan	1 μmol
	Angiostamp^TM^	0.1 μmol
Visionsense^TM^ VS3 Iridum—halogen light	IRDye-800	1 nmol
	ICG	1 nmol
	FAP-Cyan	1 nmol
	Angiostamp^TM^	1 nmol
IC-Flow^TM^—DARK	IRDye-800	0.1 μmol
	ICG	10 nano
	FAP-Cyan	0.1 μmol
	Angiostamp^TM^	0.1 μmol
IC-Flow^TM^—Room light	IRDye-800	0.1 μmol
	ICG	0.1 μmol
	FAP-Cyan	0.1 μmol
	Angiostamp^TM^	0.1 μmol
IC-Flow^TM^—LED light	IRDye-800	1 μmol
	ICG	0.1 μmol
	FAP-Cyan	0.1 μmol
	Angiostamp^TM^	1 μmol
IC-Flow^TM^—halogen light	IRDye-800	1 μmol
	ICG	0.1 μmol
	FAP-Cyan	0.1 μmol
	Angiostamp^TM^	0.1 μmol

**Table 2 animals-13-02363-t002:** Median signal-to-background ratio (SBR) recorded for Visionsense^TM^ VS3 Iridium and IC-Flow^TM^ under varying lighting conditions.

	IC Flow		
Lighting	Mean SBR	Standard Deviation	Range
Dark	5.19	6.99	0.91–19.8
Room Light	4.88	6.52	0.9–19.41
LED	1.79	1.2	0.77–4.48
Older halogen	1.99	1.34	0.86–4.98
	**Visionsense^TM^ VS3 Iridium**		
**Lighting**	**Mean SBR**	**Standard Deviation**	**Range**
Dark	40.69	74.19	0.85–221.6
Room Light	24.69	63.74	1–253.85
LED	19.82	62.06	0.86–253.69
Older halogen	13.67	41.05	0.7–212.85

## Data Availability

The data presented in the study are available upon request to the corresponding authors.

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
