# Peer review of "Impact of Surgical Lights on the Performance of Fluorescence-Guided Surgery Systems: A Pilot Study"

_animals, 2023, doi:10.3390/ani13142363_

Round 1
Reviewer 1 Report
The manuscript by Lavinia E. Chiti et al. describes the impact of surgical lights on the performances of fluorescence of 2 guided surgery systems using 4 different fluorophores, two non-targeted (ICG, IRDye-800) and two targeted (Angiostamp, FAP-Cyan).
The subject is of importance for the development of FGS and its correct use by surgeons. Globally, the experiments are well designed and the conclusions are coherent with the results. However, as indicated by the authors in the title, it is a pilot study. The experimental conditions (choice of the FGS systems) are limited and the images were acquired only on for each phantom.
In conclusion, the potential interest of this manuscript is to consider it as a whistleblower. More systematic studies must be done to draw final conclusions on this interesting subject.
Author Response
We are very grateful to the reviewer for the comments. We indeed agree that our study, due to its intrinsic limitations, must be considered as a pilot study with the objective of stimulating further investigations on this topic. A sentence has been had to further specify this concept at line 327-332
Reviewer 2 Report
This is a useful study for those using or considering the use of FGS under various lighting conditions. Some minor issues with readability and presentations are below:
Figure 1. It is difficult to associate the panels A and B with panel C. Please add arrows to panel C to identify where the dilutions are - it is confusing as to what the pinkish printed circles represent and where the samples are located.
Table 2. There is inconsistencies in the bold heading. Please make all headings bold including Visionsense Also, IC Flow and Visionsense should be centered.
The paragraph starting on line 179 should have a more obviously parallel construction to the one starting on line 171.
Line 10: "fluorophore" should be changed to "fluorophores"
Line 22: delete "the information about"
Line 33: Extra space in mSBR
Line 37: delete "the information about"
Line 309: "conclusions" should be singular, delete "with the present study"
Line 310: change "with" to "under"
Line 312: delete "the information about"
Line 85: Has the first use of FAP-Cyan as an abbreviation , but it is only written out in line 96
Author Response
We thank the reviewer for the useful comments.
Figure 1 has been modified as suggested.
The headings of Table 2 have been modified as suggested.
The two mentioned paragraphs had already a parallel construction, but a paragraph between line 205 and line 206 was missing; now it has been added and the construction of the paragraphs is equal. The only difference is that in the paragraph on IC-Flow, there is no specific information on ICG, due to the fact that with this camera system this fluorophore showed the exact same trend of the others, hence we believe it does not warrant a separate paragraph as it has for Visionsense system.
The comments line by line have been addressed in the text (lines: 10, 22, 33, 37, 85, 341, 342, 349)
Reviewer 3 Report
In this manuscript, the authors assessed the effectiveness of two fluorescence camera systems, namely IC-FlowTM and VisionsenseTM VS3 Iridum, in detecting two non-targeted fluorophores and two targeted fluorophores in various room light conditions. These conditions encompassed ambient light, new generation LED lighting, and halogen artificial light sources commonly employed in operating theaters. In terms of the tested dyes, optimal values for LOD and mSBR were achieved in dark conditions, with reasonable results also observed under room light conditions. However, the diagnostic performance of both camera systems was negatively affected by the presence of LED and halogen lights.
1. Authors should illustrate more on how did they calculated the limit of detection in 3.1 section.
2. In section 3.1, they discussed about the results in detail. However, they should also summarize them in the end and talk about the main conclusion they got from the experiments.
3. Similarly, they should also add how they calculate the SBR ratio in section 3.2.
Author Response
We thank the reviewer for the useful suggestions.
Detailed information, including references, on how we calculated LOD and SBR is reported in the material and method section (line 155-162). We believe that adding the same information in the result section would be repetitive.
The conclusion has been implemented as suggested (lines 343-348)